# Leisure Noise Exposure and Associated Health-Risk Behavior in Adolescents: An Explanatory Study among Two Different Educational Programs in Flanders

**DOI:** 10.3390/ijerph18158033

**Published:** 2021-07-29

**Authors:** Sofie Degeest, Hannah Keppler, Bart Vinck

**Affiliations:** 1Department of Rehabilitation Sciences, Ghent University, 9000 Ghent, Belgium; Hannah.Keppler@UGent.be (H.K.); Bart.Vinck@UGent.be (B.V.); 2Department of Otorhinolaryngology, Ghent University Hospital, 9000 Ghent, Belgium

**Keywords:** leisure noise, attitude, hearing protection, adolescents

## Abstract

Adolescents frequently engage in noisy leisure activities which can result in hearing-related problems. However, the effect of leisure noise exposure is liable to an individual’s risk-taking behavior. Identifying leisure noise activities and relevant factors related to risk-taking behavior in adolescents, are important to optimize hearing conservation programs targeting youngsters. The purpose of the study was to explore the presence of hearing-related symptoms, as well as noise exposure during various activities, and the use of hearing protector devices (HPDs) in adolescents in two educational programs in Flanders. In addition, their attitudes and beliefs towards noise, hearing loss, and HPDs were investigated. The final sample consisted of 247 adolescents. The most important hearing-related symptoms after noise exposure were tinnitus and noise sensitivity. With regard to leisure noise exposure, listening to PMPs was most frequently reported. The use of HPDs during most noisy activities was limited, in accordance with the presence of hearing-related symptoms, considering noise as unproblematic, and having worse scores on subscales of the beliefs about hearing protection and hearing loss scale. In the future, hearing conservation programs should target adolescents specifically for a more health-orientated behavior towards noise exposure, hearing loss, and HPDs.

## 1. Introduction

In 2015, the World Health Organization estimated that 1.1 billion young people worldwide could be at risk of developing hearing loss due to unsafe listening practices [1]. The latter entails long-term, repeated exposure to loud noise causing noise-induced hearing loss (NIHL) or hearing-related symptoms such as decreased speech understanding in unfavorable listening conditions, tinnitus, and noise sensitivity [2]. Tinnitus is the perception of a meaningless sound in the absence of an external sound source [3], while noise sensitivity refers to the increased perception and reaction to sound [4]. Leisure noise exposure in children and adolescents requires special focus [5] as hearing loss impacts speech and language development [6], psychosocial development [7], and education performance [8]. Besides hearing loss, tinnitus in children impacts emotional wellbeing [9]. 

Children and adolescents frequently engage in noisy leisure activities, such as using personal music players (PMPs) through headphones or earbuds including mp3-players, mobile phones, tablets, etc., attending discotheques and concerts, or playing musical instruments, alone or in a band or orchestra [10,11,12,13]. Hence, the effects of leisure noise exposure on the auditory system in children and adolescents should take into account the accumulated lifetime noise exposure during participation in multiple leisure activities [14]. 

In young adults, no significant differences in their hearing were found between subjects with different lifetime equivalent noise exposures based on self-report [15,16,17]. However, noisy leisure activities were associated with hearing loss in more susceptible adolescents [18], and with tinnitus in adolescents [4,10,19,20]. Although the (increase in) prevalence of NIHL due to leisure noise exposure in youngsters is not clear [19,21], the association of leisure noise exposure with hearing-related symptoms warrants caution. 

In addition, the effects of leisure noise exposure on the auditory system are liable to an individual’s risk-taking behavior. According to Widen, this behavior is based on aspects from the Theory of Planned Behavior [22] and Health Belief Model [23], in addition to risk perception implying an individual’s awareness of the risks of leisure noise exposure [24]. More specifically, attitudes, subjective norms, and perceived behavioral control from the Theory of Planned Behavior [22], and perceived benefits and barriers, as well as triggers to action from the Health Belief Model [23], are taken into account. Attitudes refer to an individual’s valuation towards a certain phenomenon, while subjective norms indicate whether or not an individual’s engagement in a particular behavior is influenced by others. The perceived behavioral control comprises the control beliefs promoting or inhibiting the specific behavior. Perceived benefits and barriers refer to the belief that a more health-orientated behavior reduces the threat or coincides with obstacles, respectively. Finally, triggers are factors that may lead to behavioral change [22,23,24]. 

In young adults, it was found that their attitudes and beliefs regarding noise, hearing loss, and hearing protector devices (HPDs), had a significant impact on hearing [25], as well as on the presence of tinnitus [26]. This was also reported in adolescents [27,28]. Further, factors such as age, gender, cultural difference, and socio-economic status can influence attitudes and beliefs towards noise exposure, hearing loss, and HPDs [29,30,31]. In Flanders, the upper part of Belgium, there are different educational programs, such as the general secondary program (GSP) (in Dutch: algemeen secundair onderwijs) and technical secondary program (TSP) (in Dutch: technisch secundair onderwijs) The former consists of a broad general education that enables students to follow higher education, while the latter can be described as a general and technical-theoretical education, often supplemented with practical lessons, that enables students to follow higher (technical) education or to exercise a specific profession. Although various learning environments and activities in both general and technical secondary school settings can include high-intensity levels, for example, music or gym classes (e.g., [32]), the TSP can also include technical classes involving higher intensity levels (e.g., woodworking). Identifying leisure noise activities and relevant factors related to risk-taking behavior in adolescents, are important to optimize hearing conservation programs targeting youngsters. Those hearing conservation programs aim for a more health-orientated behavior by providing information regarding the effects of hearing loss, increasing the awareness of the risk of excessive noise exposure, and knowledge concerning the availability and use of HPDs [33,34]. 

The aim of the study was to explore the presence of hearing-related symptoms, as well as attendance and estimation of loudness during various leisure noise activities, and the use of HPDs in adolescents in two different educational programs in Flanders. In addition, their attitudes and beliefs towards noise, hearing loss, and HPDs were investigated. We hypothesized that tinnitus and noise sensitivity after noise exposure would be reported frequently, that adolescents would participate in various leisure noise activities without using HPDs, and that their attitudes and beliefs towards noise, hearing loss, and HPDs would be positive. We also hypothesized differences in the presence of hearing-related symptoms, and beliefs towards noise, hearing loss, and HPDs between adolescents from two educational programs. 

## 2. Materials and Methods

### 2.1. Study Sample

This study was a cross-sectional study using a self-administered questionnaire in a group of high school students from the third grade of secondary education. The principals of four chosen high schools were contacted and asked to participate in the study. These high schools were situated in East-Flanders, which is a province in Flanders.

The final sample consisted of 247 high school students (164 females and 83 males), with an age range of 15 to 20 years (mean 17.5 years, SD 1.36). Furthermore, 105 (42.5%, of which 78 females and 27 males) and 142 (57.5%, of which 87 females and 55 males) students participated in the GSP and TSP, respectively. 

The study was approved by the Ethical Committee of Ghent University Hospital and was conducted in accordance with the ethical standards stipulated in the Helsinki declaration for research involving human subjects (BC-07045 and 29 April 2019).

### 2.2. Questionnaire

As the evaluation of noise exposure in young people should contain different aspects such as the evaluation of the amount of noise exposure, hearing loss, and hearing-related symptoms, as well as attitudes towards noise exposure, hearing loss, and use of HPDs, a Dutch questionnaire encompassing these factors, was needed. However, a questionnaire addressing all of these aspects through one questionnaire was lacking. Hence, Keppler et al. [35] designed a questionnaire based on available literature regarding noise exposure and the assessment of noise-induced tinnitus and noise-induced hearing loss [16,26,29,36,37]. The translated preliminary version of the questionnaire was pretested by a semi-structured interview-based assessment on a group of young adults not included in the current study. The results of the interviews were analyzed based on the frequency distribution of the given answers, the comprehensibility of the questions, answers, and instructions [35], as well as the internal consistency using Cronbach’s alpha and reliability measures [35,38]. As a result, the questionnaire was refined in order to be useful in further research.

The final questionnaire had four sections. The first section included several socio-demographic variables such as age and gender. The second section consisted of questions pertaining to the medical history of ear-related disorders, the subjective hearing status, problems with speech understanding, and the presence of hearing-related symptoms (i.e., the presence of tinnitus, dullness, ear pain, and noise sensitivity). In the case of tinnitus, the questions were formulated as such that it would be possible to distinguish between temporary and chronic tinnitus, and that subjects indicating to have chronic tinnitus are subjects with continuous or intermittent tinnitus. 

In the third section of the questionnaire, participant’s participation in noisy activities, as well as their estimation of loudness in terms of communicative effort, was administered for several leisure activities that are common among young people such as visiting nightclubs, using PMPs, and playing musical instruments. Five levels of loudness were considered: (1) level of a normal conversation, (2) level of a loud conversation, (3) level at which one must shout over one meter in order to be heard (e.g., over a table), (4) level at which one must shout over a near distance in order to be heard (e.g., someone less than an arm’s length away), (5) level that makes communication impossible [33]. Furthermore, participants were asked about their use of HPDs during the activities on the one hand, and the advantages and disadvantages pertaining to wearing HPDs, as well as their willingness to wear HPDs on the other hand.

The fourth section consisted of a Dutch modified version of the ‘Youth Attitude to Noise Scale’ (YANS) [29,35] and a Dutch modified version of the ‘Beliefs about Hearing Protection and Hearing Loss’ (BAHPHL) [35,37]. The YANS evaluates a subject’s attitude towards noise and consisted of 19 items that were measured using a five-point Likert scale ranging from ‘totally disagree’ to ‘totally agree’. A higher score on the YANS indicates a positive or pro-noise attitude representing an attitude where noise is seen as unproblematic. The 19 items are divided over four factors representing attitudes towards noise associated with elements of youth culture (factor 1: 8 items), the ability to concentrate in noisy environments (factor 2: 3 items), daily noises (factor 3: 4 items), and intent to influence the sound environment (factor 4: 4 items) [29]. The BAHPHL instrument evaluates the attitudes towards hearing loss and HPDs and contains 24 items which can be divided into seven factors: susceptibility to hearing loss (factor 1: 6 items), the severity of consequences of hearing loss (factor 2: 3 items), benefits of preventive action (factor 3: 3 items), barriers to preventive action (factor 4: 4 items), behavioral intentions (factor 5: 3 items), social norms (factor 6: 2 items), and self-efficacy (factor 7: 3 items) [37]. Consistent with the YANS, the items were evaluated by a five-point Likert scale with higher scores corresponding to a more positive attitude, meaning that one does not care about the possible consequences of hearing loss and is unaware of the benefits of wearing HPDs. 

To ensure that the questionnaire was completed correctly by the subjects, instructions were provided at the beginning of the form as well as for each new section. All terminology regarding leisure noise, hearing, and tinnitus, as well as HPDs, was explained and appropriate examples were given.

### 2.3. Statistical Analysis

Statistical analysis was performed using SPSS version 25 (SPSS Inc., Chicago, IL, USA). Descriptive parameters and normality analyses were established for the different questionnaire outcomes. 

Subsequently, simple analyses were performed to evaluate the relation of different variables with the educational program (i.e., students from the GSP and TSP). In the case of continuous variables, an independent samples *t*-test was conducted. To examine possible correlations between categorical variables, chi-square tests (2 × 2 or 2 × 3 tables) were performed. When the chi-squared test was significant, pairwise comparisons were performed for the 2 × 3 tables, using Bonferroni corrections of the *p*-values (α = 0.05/3). If one or more cells had an expected count of less than five, Fisher’s exact test was used. Except after Bonferroni corrections, *p*-values less than 0.05 were used indicating statistical significance.

## 3. Results

### 3.1. Hearing-Related Symptoms

Table 1 provides an overview of the presence of hearing-related symptoms after exposure to noise. In general, one of the most common reported symptoms among the students was the experience of tinnitus, whereby 64.8% of the total amount of students reported to have experienced tinnitus at least once after exposure to noise. Considering the students from the GSP and TSP separately, tinnitus was reported significantly more by the students from the GSP (76.2%) compared to the TSP (56.3%), χ^2^ = 10.427; *p* < 0.05. In the majority of all students (97.5%), tinnitus was temporary and disappeared within 72 h, whereas 2.5% of the students reported chronic tinnitus. Although the difference in the occurrence of temporary and chronic tinnitus between the students from the GSP and TSP was not statistically significant (Fisher’s exact test = 4.103; *p* > 0.05), it should be noted that the students reporting chronic tinnitus were all from the TSP group. For the remaining hearing-related symptoms, no significant differences were found between the students from the GSP and TSP (chi-squared tests, *p* > 0.05). 

In addition, difficulties with speech understanding were questioned among the students. Overall, the majority of the students reported difficulties with speech understanding ‘sometimes’ for each of the different listening situations (Table 1). Besides, respectively 25.7% and 21.1% of the students from the GSP and TSP group reported difficulties with speech understanding ‘always’ for the noisy listening situations. No significant differences were found between the subjects from the GSP and TSP (chi-squared tests, *p* > 0.05).

### 3.2. Leisure Noise Exposure and the Use of HPDs

The students included in this study participated in a variety of activities. The highest attendance was reported for listening to PMPs through headphones or earbuds (98.4%), watching movies or plays (97.5%), listening to music through loudspeakers (89.1%), and visiting or working at nightclubs or music venues (82.3%). Furthermore, visiting and working at nightclubs and music venues were described as the loudest, where one must shout over a near distance. Participation and self-estimated median loudness for the different activities were compared between the students from the GSP and TSP (Table 2). For each of the activities, no significant differences were found between the students from the GSP and TSP pertaining to their participation in the different activities (chi-squared tests, *p* > 0.05) as well as their estimated subjective loudness of the activities (independent samples *t*-tests, *p* > 0.05). Although not statistically significant, descriptive data suggests that students from the TSP reported more participation in occupational noise activities and the use of noisy tools compared to the students from the GSP.

In addition, the subjects were asked about their use of HPDs during the activities they participate in. For all of the activities, the majority of students did not wear HPDs. The highest grades of wearing HPDs were reported for using noisy tools (always: 25.5%, sometimes: 22.0%), attending or working at musical concerts (always: 11.3%, sometimes: 24.7%), and visiting or working at nightclubs of music venues (always: 7.1%, sometimes: 19.6%). No significant difference in the use of HPDs during the different activities was found between the students from the GSP and TSP (chi-squared tests, *p* > 0.05) (Table 2). Regarding the subjects wearing HPDs always or sometimes, the majority (64.3%) wore foam earplugs, while 23.8% indicated wearing universal earplugs. The remaining subjects wore earmuffs (8.7%) or custom-made earplugs (3.2%). Additionally, no significant difference in the type of HPDs that was used was found between the students from the GSP and TSP (Fisher exact test, *p* > 0.05). Furthermore, of subjects who indicated wearing HPDs in one or more activities, the majority indicated concerns about their hearing as the main reason to wear HPDs (47.6%). Besides, 30.7% of the subjects indicated that loud music is the main reason to wear HPDs, while 4.8% wear HPDs because of a detected hearing loss. The remaining subjects (16.9%) indicated no clear reason for wearing HPDs. Reasons for wearing HPDs were also not significantly different between the students from the GSP and TSP (Fisher’s exact test = 4.417; *p* > 0.05).

Finally, all subjects were asked whether they were already recommended to use HPDs as well as about the disadvantages and their willingness of wearing HPDs (Table 3). The majority of students from both the GSP (72.8%) and TSP (73.8%) had already been recommended to wear hearing protection. Specifically, the majority of these students, both within the GSP (78.7%) and TSP (64.4%), were advised by their parents to wear hearing protection. Considering the disadvantages of wearing HPDs, the main reasons not to wear HPDs were related to discomfort, reduced quality of the music, and difficulties with speech understanding, as well as the opinion that wearing hearing protection, was useless. According to chi-squared tests, no significant differences were found between the students from the GSP and TSP (*p* > 0.05). In contrast, experiencing hearing loss and compulsory wearing of HPDs were the main reasons to be willing to wear HPDs, whereby no differences were found between the students from the GSP and TSP (χ^2^ = 1.867; *p* > 0.05).

### 3.3. Attitudes towards Noise, Hearing Loss, and HPDs

Table 4 reflects the mean and standard deviations of the scores on the YANS and BAHPHL. Concerning the subscales of the YANS, the highest average score was found for the attitudes regarding daily noise, whereas the lowest average score was related to the attitudes intending to influence the sound environment. The score on the entire YANS did not show any statistical difference between the students from the GSP and TSP (t(240) = −1.264; *p* > 0.05). In contrast, a significant difference between the GSP and TSP group was found for the scores on the factors related to the concertation in noisy environments (t(240) = −2.756; *p* < 0.01) as well as the intent to influence sound environment (t(240) = −2.789; *p* < 0.01). Specifically, students from the TSP had significantly higher scores on both these factors. 

For the subscales of the BAHPHL, the highest and lowest average scores were respectively found for the social norms and the severity of consequences of hearing loss. No significant differences in the scores were found between the students from the GSP and TSP (independent samples *t*-test, *p* > 0.05), except for the factor related to the severity of the consequences of hearing loss (t(234) = −2.893; *p* < 0.01). Specifically, students from the TSP showed significantly higher scores for this factor, which indicates that students from the TSP were less aware of the severity of hearing loss compared to the students from the GSP.

## 4. Discussion

The present study explored hearing-related symptoms, leisure noise exposure, and attitudes and beliefs towards noise, hearing loss, and HPDs in 247 adolescents between 15 and 20 years in two educational programs in Flanders using a self-report questionnaire. 

With regard to hearing-related symptoms after noise exposure, the presence of temporary and chronic tinnitus in the current study was 63.2% and 1.6%, respectively. Generally, this is lower as compared to previous studies [4,26,29,39,40,41,42]. Amongst others, differences in the definition and questioning of the symptoms can explain the variation in the prevalence of temporary and chronic tinnitus. However, using identical methods, temporary and chronic tinnitus in Flemish young adults between 18 and 30 years was 68.5% and 6.4%, respectively [26]. Furthermore, the presence of temporary and chronic tinnitus in university students was higher as compared to adolescents in Flanders [41,42]. In addition to age, female gender, socio-economic status, attitudes, and noise exposure can be related to the occurrence of noise-induced tinnitus [20,27,28,40,43]. In the current study, another factor, educational program, influenced the presence of tinnitus. More specifically, significantly more adolescents in GSP reported tinnitus as compared to those in TSP. Besides tinnitus, noise sensitivity after noise exposure was frequently reported by the adolescents. As both hearing-related symptoms are triggers to action [24], they could be used in hearing conservation programs targeting adolescents. 

The most frequent leisure noise activities reported by the adolescents were listening to PMPs through headphones or earbuds, watching movies or plays, listening to music through loudspeakers, and visiting or working at nightclubs or music venues. As compared to young adults, the use of PMPs is higher in adolescents, consistent with previous studies [15,16,44,45,46]. Hence, it is important not only considering the accumulation of leisure noise activities but also taking into account listening habits that tend to change during a lifetime. In hearing conservation programs targeting adolescents specifically the use and acoustic coupling of PMPs should be considered [47], with recommendations regarding listening levels during PMP use, and the use of isolating earphones in noisy environments [48]. 

Regarding risk-taking behavior, attitudes and beliefs towards noise exposure, hearing loss, and HPDs were questioned using the YANS and BAHPHL in the current study. These scales evaluate the factors from the Theory of Planned Behavior [22] and Health Belief Model [23], according to the framework regarding leisure noise exposure provided by Widen [24]. In the current study, adolescents on average report noise more as unproblematic. Moreover, the severity of consequences of hearing loss is on average more negatively assessed, while in contrast, the barriers to preventive action, behavioral intentions, and social norms are on average more positively evaluated. Further, the attitudes and beliefs towards noise exposure, hearing loss, and HPDs were mostly worse as compared to young adults [16,25,41], and were more in line with the scores for adolescents previously reported [29,42]. Besides age, and other confounding variables such as gender, cultural difference, and socio-economic status [29,30,31], it was hypothesized that information and knowledge can explain variation between studies. In the current study, adolescents in TSP had significantly worse scores regarding attitudes towards noise exposure, and the subscale severity of consequences of hearing loss of the BAHPHL scale as compared to adolescents in GSP. As the former also report less tinnitus, they might thus be less aware of the reduced communication skills associated with hearing loss, and hearing-related symptoms after noise exposure. 

In addition to questioning attitudes and beliefs towards noise exposure, hearing loss, and HPDs in relation to confounding variables, these aspects were previously also related to hearing status. More specifically, young adults with more problematic attitudes regarding noise exposure or positively evaluating barriers to preventive action, have already significant hearing damage in comparison to those with more negative or neutral attitudes and beliefs [25]. Finally, the importance of questioning attitudes and beliefs towards noise exposure, hearing loss, and HPDs lies within the evaluation of the effectiveness of preventive campaigns [49,50]. 

The reported use of HPDs during most noisy exposure activities was limited in the current study. However, during some activities, i.e., using noisy tools, attending or working at musical concerts, festivals, nightclubs, or music venues, adolescents reported wearing HPD more as compared to previous studies in young adults [16] and adolescents [42]. Previous research indicated that HPD usage was significantly correlated with hearing-related symptoms, barriers, and norms [16,24,25]. Thus, subjects without hearing-related symptoms, and more positively evaluating barriers to preventive action and social norms, are less likely to protect their hearing. Moreover, the barriers to wearing HPDs, i.e., comfort, impact on sound quality and speech understanding, and usefulness, are consistent with findings in young adults [51,52]. Manufacturers of HPDs could target these groups specifically, e.g., regarding looks, design, marketing, and packaging of the HPDs [52]. 

One of the strategies to prevent damage to the auditory system by leisure noise exposure in youngsters is controlling noise levels. In Flanders, there is noise legislation for indoor and outdoor music venues, although the compliance should be increased [53]. However, other leisure noise activities should also be considered; e.g., using PMPs [54]. Further, it is recommended to limit leisure noise exposure in children at a more stringent 8-h L_EX_ of 80 dBA [55]. In addition, at an individual level, achieving a more health-orientated behavior should be aimed for [33,34]. During adolescence, noise exposure during leisure noise activities is increasing, and attitudes and beliefs are formed. It is therefore important to implement hearing conservation programs before the age of onset of hearing-related problems due to noise exposure, taking into account the diversity of adolescents in different educational programs. Discussing and learning to estimate hazardous noise and listening levels, noise-induced hearing-related symptoms, barriers against the use of HPDs, as well as correct insertion of HPDs are important in hearing conservation programs targeting adolescents. The current study also revealed that the majority of the adolescents have been recommended to wear HPDs, mostly by their parents. Their role in preventive strategies, e.g., parents should wear HPDs so children can adhere to proper behavior [55], should hence not be neglected. 

The results of the current study should be considered taking into account some limitations. First, the sample might not be representative of all Flemish adolescents, with regard to geographic distribution and educational programs. Nevertheless, it was ensured that two educational programs within a region in Flanders were represented. Second, a self-report questionnaire was used, in which noise exposure was estimated. Although the questionnaire considers different leisure activities in detail, it is possible that the time spent on leisure activities and the estimation of loudness are imprecise. Other measures, not restricted to prespecified activities during specific periods in life, such as the Noise Exposure Structured Interview [56], might be more accurate in the estimation of lifetime noise exposure. Finally, the results of the current study could be strengthened with measurements of hearing status, such as pure tone audiometry including extended high frequencies, speech audiometry in noise, otoacoustic emissions, and electrophysiological measurements for the assessment of cochlear synaptopathy [57]. Further research including a larger population of adolescents from different educational programs, combining subjective, behavioral, and objective measures in a longitudinal study design is necessary to further investigate the effects of leisure noise exposure on the auditory system, and attitudes and beliefs regarding noise exposure, hearing losss and HPDs. In the future, this could optimize hearing conservation programs and prevent hearing loss and hearing-related symptoms in youngsters. 

## 5. Conclusions 

The present study of 247 adolescents between 15 and 20 years in two educational programs in Flanders showed the presence of hearing-related symptoms such as temporary and chronic tinnitus and noise sensitivity after noise exposure. With regard to leisure noise exposure, in this population, listening to PMPs was more frequently reported, as compared to attending nightclubs or music venues. The use of HPDs during most noise exposure activities was limited, in accordance with the presence of hearing-related symptoms, considering noise as unproblematic, and having worse scores on subscales of the BAHPHL related to barriers to preventive action, behavioral intentions, and social norms. Between the adolescents in the two educational programs, differences in the presence of tinnitus after noise exposure, and subscales of the YANS and BAHPHL regarding concentration in noisy environments, the intention to influence the sound environment, and severity of the consequences of hearing loss were found. In the future, hearing conservation programs should target adolescents specifically for a more health-orientated behavior towards noise exposure, hearing loss and HPDs. 

## Figures and Tables

**Table 1 ijerph-18-08033-t001:** Overview of hearing-related symptoms and speech understanding difficulties for the total sample as well as distributed for the students from the general secondary program (GSP) and technical secondary program (TSP).

Variable		Total Sample % (*n*)	Educational Program	
			GSP % (*n*)	TSP % (*n*)
**Hearing-related symptoms after noise exposure**				
Subjective hearing loss	Total *	100.0 (243)	42.4 (103)	57.6 (140)
Always		14.8 (36)	19.4 (20)	11.4 (16)
Sometimes		56.0 (136)	52.4 (54)	58.6 (82)
Never		29.2 (71)	28.2 (29)	30.0 (42)
Dullness	Total ^†^	100.0 (244)	42.6 (104)	57.4 (140)
Always		24.6 (60)	27.9 (29)	22.1 (31)
Sometimes		51.2 (125)	52.9 (55)	50.0 (70)
Never		24.2 (59)	19.2 (20)	27.9 (39)
Ear pain	Total ^‡^	100.0 (243)	42.8 (104)	57.2 (139)
Always		11.5 (28)	11.5 (12)	11.5 (16)
Sometimes		41.2 (100)	42.3 (44)	40.3 (56)
Never		47.3 (115)	46.2 (48)	48.2 (67)
Noise sensitivity	Total ^¥^	100.0 (241)	42.8 (105)	57.2 (136)
Always		16.2 (39)	13.3 (14)	18.4 (25)
Sometimes		66.4 (160)	75.2 (79)	59.5 (81)
Never		17.4 (42)	11.5 (12)	22.1 (30)
Tinnitus	Total	100.0 (247)	42.5 (105)	57.5 (142)
Yes		64.8 (160)	76.2 (80)	56.3 (80)
No		35.2 (87)	23.8 (25)	43.7 (62)
Tinnitus duration	Total	100.0 (160)	50.0 (80)	50.0 (80)
Temporary (<72 h)		97.5 (156)	100.0 (80)	95.0 (76)
Chronic		2.5 (4)	0.0 (0)	5.0 (4)
**Speech understanding difficulties**				
Speech understanding in quiet with several persons	Total	100.0 (247)	42.5 (105)	57.5 (142)
Always		12.1 (30)	12.4 (13)	12.0 (17)
Sometimes		76.5 (189)	79.0 (83)	74.6 (106)
Never		11.3 (28)	8.6 (9)	13.4 (19)
Speech understanding in noise	Total	100.0 (247)	42.5 (105)	57.5 (142)
Always		12.1 (30)	25.7 (27)	21.1 (30)
Sometimes		76.5 (189)	70.5 (74)	71.2 (101)
Never		11.3 (28)	3.8 (4)	7.7 (11)
Speech understanding during a telephone conversation	Total ^§^	100.0 (246)	42.3 (104)	57.7 (142)
Always		7.3 (18)	6.7 (7)	7.7 (11)
Sometimes		70.7 (174)	74.0 (77)	68.4 (97)
Never		22.0 (54)	19.3 (20)	23.9 (34)

Note: GSP, general secondary program; TSP, technical secondary program. * 2 missing values in the GSP group and 2 missing values in the TSP group; ^†^ 1 missing value in the GSP group and 2 missing values in the TSP group; ^‡^ 1 missing value in the GSP group and 3 missing values in the TSP group; ^¥^ 6 missing values in the TSP group; ^§^ 1 missing value in the GSP group.

**Table 2 ijerph-18-08033-t002:** Percentage of student’s attendance in each activity as well as the median loudness, and the percentage of subjects wearing hearing protector devices (HPDs), distributed for the students from the GSP (*n* = 105) and TSP (*n* = 142).

Activity	Attendance (%)			Subjective Loudness		Wearing HPDs (%)					
						Always		Sometimes		Never	
	Total	GSP	TSP	GSP	TSP	GSP	TSP	GSP	TSP	GSP	TSP
Listening to PMPs through headphones or earbuds	98.4	98.1	98.6	Shout over 1 m		n/a	n/a	n/a	n/a	n/a	n/a
Watching movies or plays	97.5	98.1	97.1	Shout over 1 m		1.0	0.0	2.1	0.9	96.9	99.1
Listening to music through loudspeakers	89.1	91.3	87.4	Loud conversation	Shout over 1 m	0.0	1.0	3.4	4.9	96.6	94.1
Visiting or working at nightclubs or music venues	82.3	81.7	82.7	Shout over near distance		6.1	7.8	17.1	21.6	76.8	70.6
Attending or working at musical concerts or festivals	65.7	70.2	62.3	Shout over near distance		8.6	13.8	28.5	21.2	62.9	65.0
Attending or participating in sport events	47.7	45.6	49.3	Loud conversation	Shout over 1 m	2.2	0.0	2.2	1.7	95.6	98.3
Gaming with headphones or earbuds	30.5	24.0	35.3	Shout over 1 m	Loud conversation	n/a	n/a	n/a	n/a	n/a	n/a
Practicing a musical instrument	28.3	29.8	27.1	Loud conversation		0.0	2.7	0.0	8.1	100.0	89.2
Using noisy tools	26.9	24.0	29.0	Shout over 1 m		25.0	25.7	29.2	17.2	45.8	57.1
Gaming with loudspeakers	20.7	16.3	23.9	Loud conversation		0.0	0.0	0.0	3.2	100.0	96.8
Occupational noise	15.4	10.7	19.0	Shout over 1 m		10.0	0.0	10.0	9.1	80.0	90.9
Other noisy leisure-time activities	13.0	11.7	14.1	Shout over 1 m		8.3	5.9	8.3	0.0	83.4	94.1
Playing in a band or orchestra	7.0	11.5	3.6	Shout over 1 m		0.0	0.0	9.1	20.0	90.9	80.0

Note: GSP, general secondary program; TSP, technical secondary program; n/a, not applicable.

**Table 3 ijerph-18-08033-t003:** Recommendation to wear HPDs, as well as reasons not to use HPDs and the willingness to use HPDs for the total sample as well as distributed for the students from the GSP and TSP.

Variable		Total Sample % (*n*)	Educational Program	
			GSP % (*n*)	TSP % (*n*)
**Recommended to wear HPDs**	Total *	100.0 (244)	42.2 (103)	57.8 (141)
Yes		73.4 (179)	72.8 (75)	73.8 (104)
No		26.6 (65)	27.2 (28)	26.2 (37)
**Reasons not to use HPDs**	Total ^†^	100.0 (232)	40.9 (95)	59.1 (137)
HPDs are not useful				
Agree		31.5 (73)	34.7 (33)	29.2 (40)
Not agree		68.5 (159)	65.3 (62)	70.8 (97)
HPDs are too expensive				
Agree		7.3 (17)	7.4 (7)	7.3 (10)
Not agree		92.7 (215)	92.6 (88)	92.7 (127)
HPDs are not comfortable				
Agree		63.8 (148)	68.4 (65)	60.6 (83)
Not agree		36.2 (84)	31.6 (30)	39.4 (54)
HPDs are not cool				
Agree		14.2 (33)	11.6 (11)	16.1 (22)
Not agree		85.8 (199)	88.4 (84)	83.9 (115)
HPDs hinder hearing the music				
Agree		37.9 (88)	43.2 (41)	34.3 (47)
Not agree		62.1 (144)	56.8 (54)	65.7 (90)
HPDs hinder speech understanding				
Agree		42.7 (99)	47.4 (45)	39.4 (54)
Not agree		57.3 (133)	52.6 (50)	60.6 (83)
**Willingness to use HPDs**	Total ^‡^	100.0 (228)	43.4 (99)	56.6 (129)
Willing to use HPDs if they are free		14.9 (34)	4.8 (11.1)	17.8 (23)
Willing to use HPDs if diagnoses with hearing loss		45.6 (104)	46.5 (46)	45.0 (58)
Willing to use HPDs if it is obligated		39.5 (90)	42.4 (42)	37.2 (48)

Note: GSP, general secondary program; TSP, technical secondary program; HPDs, Hearing Protection Devices. * 2 missing values in the GSP group and 1 missing value in the TSP group; ^†^ 10 missing values in the GSP group and 5 missing values in the TSP group; ^‡^ 6 missing values in the GSP group and 13 missing values in the TSP group.

**Table 4 ijerph-18-08033-t004:** Description of the attitudes and beliefs towards noise, hearing loss, and HPDs for the total sample as well as distributed for the students from the GSP and TSP.

Variable		Total Sample	Educational Program	
			GSP	TSP
YANS	Total (*n*) *	242	102	140
Elements of youth culture	mean (SD) (range)	2.9 (0.64) (1.38–4.75)	3.0 (0.64) (1.38–4.75)	2.9 (0.64) (1.38–4.35)
Concentration in noisy environments		3.0 (0.86) (1.00–5.00)	2.8 (0.85) (1.00–4.67)	3.1 (0.84) (1.00–5.00)
Daily noise		3.2 (0.78) (1.00–5.00)	3.2 (0.76) (1.25–4.75)	3.2 (0.79) (1.00–5.00)
Intent to influence sound environment		2.4 (0.66) (1.00–4.00)	2.3 (0.65) (1.00–4.00)	2.5 (0.64) (1.00–4.00)
Entire YANS		2.9 (0.44) (1.58–3.95)	2.8 (0.46) (1.63–3.95)	2.9 (0.43) (1.58–3.94)
BAHPHL	Total (*n*) ^†^	236	100	136
Susceptibility to hearing loss	mean (SD) (range)	2.3 (0.61) (1.00–3.67)	2.4 (0.59) (1.00–3.67)	2.3 (0.63) (1.00–3.67)
Severity of the consequences of hearing loss		2.0 (0.73) (1.00–3.67)	1.8 (0.70) (1.00–3.33)	2.1 (0.73) (1.00–3.67)
Benefits of preventive action		2.1 (0.71) (1.00–4.00)	2.1 (0.67) (1.00–3.67)	2.1 (0.74) (1.00–4.00)
Barriers to preventive action		3.3 (0.75) (1.00–5.00)	3.4 (0.78) (1.00–5.00)	3.2 (0.73) (1.00–5.00)
Behavioral intentions		3.3 (0.91) (1.00–5.00)	3.3 (0.84) (1.33–5.00)	3.2 (0.96) (1.00–5.00)
Social norms		3.4 (0.81) (1.00–5.00)	3.5 (0.79) (1.50–5.00)	3.4 (0.82) (1.00–5.00)
Self-efficacy		2.8 (0.71) (1.00–4.67)	2.9 (0.71) (1.00–4.67)	2.7 (0.70) (1.00–4.67)

Note. GSP, general secondary program; TSP, technical secondary program; YANS, Youth Attitudes to Noise Scale; BAHPHL, Beliefs About Hearing Protection and Hearing Loss. * 3 missing values in the GSP group, and 2 missing values in the TSP group; ^†^ 5 missing values in the GSP group, and 6 missing values in the TSP group.

## Data Availability

Not applicable.

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
