# Peer review of "Leisure Noise Exposure and Associated Health-Risk Behavior in Adolescents: An Explanatory Study among Two Different Educational Programs in Flanders"

_ijerph, 2021, doi:10.3390/ijerph18158033_

Round 1
Reviewer 1 Report
- This is a manuscript which is try to Identify leisure noise activities and relevant factors related to risk-taking behavior in adolescents, are important to optimize hearing conservation programs targeting youngsters. Therefore, this study aims to evaluate in adolescents the presence of hearing-related symptoms, as well as attendance and estimation of loudness during various leisure noise activities, and use of hearing protector devices (HPDs). In addition, attitudes and beliefs towards noise, hearing loss and HPDs were investigated. In this population, the authors found that listening to personal music players was more frequently reported, as compared to attending nightclubs or music venues. The use of HPDs during most noise exposure activities was limited, in accordance with the presence of hearing-related symptoms, considering noise as unproblematic, and having worse scores on subscales of the BAHPHL questionnaire related to barriers to preventive action, behavioral intentions and social norms.
- Besides, this study was a cross-sectional study using a self-administered questionnaire in a group of high school students from the third grade of secondary education. In section of “Results”, the authors stated that “In general, one of the most common reported symptoms among the students was the experience of tinnitus, whereby 64.8% of the total amount of students reported to have experienced tinnitus at least once after exposure to noise…..In the majority of all students (97.5%), tinnitus was temporary and disappeared within 72 hours, whereas 2.5% of the students reported chronic tinnitus.” As you know, tinnitus can be categorized as continuous, intermittent or temporary. Both continuous and intermittent tinnitus are chronic conditions, whereas an acute episode of tinnitus that does not recur is considered to be temporary. In intermittent tinnitus, periods of presence and absence of the tinnitus alternate, and occur in a more or less periodic fashion. This contrasts with patients who perceive tinnitus constantly. However, it could be possible for the participants to be confused by the questionnaire. Could you please provide more data in detail or elaborate this point further?
- In section of “Discussion”, the authors said that “It is therefore important to implement hearing conservation programs before the age of onset of hearing-related problems due to noise exposure, taking into account the diversity of adolescents in different educational programs. Discussing hazardous noise and listening levels, noise-induced hearing-related symptoms, barriers against the use of HPDs, as well as correct insertion of HPDs are important in hearing conservation programs targeting adolescents.” As you know, health Promotion is the process of enabling people to increase control over, and to improve, their health. In my opinion, it should be very important for adolescents to know how to tell if they are listening to dangerous noise levels without noise dosimeter. Could you share your concepts or implementation of these healthy behaviors based on your research?
Reviewer 2 Report
,
Reviewer 3 Report
This paper about the leisure noise habits of a large sample of Belgian 15-20 year olds reads well. While the findings are similar to those reported previously on other adolescent groups, the data are presented well and I have no reservations about its publication in its present form.
Round 2
Reviewer 2 Report
Sugestions were accepted by the authors.